# Implementation of SCID Screening in Denmark

**DOI:** 10.3390/ijns7030054

**Published:** 2021-08-12

**Authors:** Marie Bækvad-Hansen, Dea Adamsen, Jonas Bybjerg-Grauholm, David Michael Hougaard

**Affiliations:** Danish Center for Neonatal Screening, Department for Congenital Disorders, Statens Serum Institut, 2300 Copenhagen, Denmark; deaadamsen3009@hotmail.com (D.A.); jogr@ssi.dk (J.B.-G.); dh@ssi.dk (D.M.H.)

**Keywords:** neonatal screening, SCID, RealtimePCR

## Abstract

Screening for SCID was added to the Danish Neonatal Screening Program in February 2020. The screening uses a RealtimePCR kit and we here present the results and experiences with the validation of the kit and the first 10 months of screening.

## 1. Introduction

The Danish Neonatal Screening Program was initiated in 1975 with screening for phenylketonuria (PKU). The screening panel has, since, expanded numerous times and now includes congenital hypothyroidism, multiple disorders of fatty acid oxidation, organic acidemias and cystic fibrosis. The complete list of disorders in the Danish neonatal screening panel can be found on the Danish neonatal screening website [1]. In February 2020, screening for Severe Combined Immunodeficiency (SCID) was added as the 18th disorder to the screening panel. The aim of the screening was to identify infants with classical SCID with complete T-cell depletion. Thus, newborns with functional SCID and normal T-cell count will not be detected by the screening. SCID is characterized by severe T-cell lymphopenia and the disorder is fatal without early treatment [2]. Screening for SCID based on the quantification of T-cell receptor excision circles (TREC) in dried blood spot (DBS) samples has been implemented in newborn screening panels worldwide [3,4]. In the Danish Neonatal Screening Program, the quantification of TREC is performed by RealtimePCR with an assay that amplifies both TREC and a two-copy reference gen, RPP30.

Prior to implementing the screening in Denmark, a large validation study, including more than 6000 archived neonatal DBS samples as well as SCID-positive DBS samples, was performed in order to evaluate the assay performance and determine the cut-off values for the screening. This paper describes the experiences and results of the first 10 months of screening for SCID in Denmark with a total of 53.221 newborn DBS samples.

## 2. Materials and Methods

### 2.1. Subjects

The validation study was performed on archived DBS from the Danish Neonatal Screening Biobank and the validation cohort was sampled from Danish newborns who had their newborn screening performed between July 2013 and September 2013.

SCID-positive DBS samples were provided by Perkin Elmer. A total of 17 SCID-positive samples were included in the study.

Validation samples were as default run in singleton, whereas SCID-positive samples were run in triplicates.

A subset of the validation study was used to assess inter and intra assay variation as part of the kit validation prior to initiating the screening.

The evaluation of the first 10 months of neonatal screening for SCID in Denmark was based on all newborns in Denmark, Greenland and the Faroe Islands, who had a newborn screening performed between 1st February 2020 and 30th November 2020. 

### 2.2. Samples

Samples were collected as capillary blood samples from heel pricks. The blood sample was transferred to filter paper, either by direct contact between the filter paper and the heel prick or by use of capillary tubes free of additives such as anticoagulants. The samples were dried and subsequently sent to Statens Serum Institut at room temperature. Sampling was preformed 48 to 72 h after birth [5]. Residual material was stored in the Danish Neonatal Screening Biobank at −20 °C [6]. 

### 2.3. Realtime PCR

The screening was based on the EONIS PCR kit (Perkin Elmer, Turku, Finland). The kit was a multiplex RealtimePCR assay with primers and TaqMan probes for TREC and a two-copy reference gene, RPP30. The kit also contained controls in three different levels—no TREC, low levels of TREC and high levels of TREC. Samples and controls were punched in 96-well plates, with one 3.2 mm dried blood spot punch in each well. DNA was extracted using the EONIS DNA extraction kit. The extraction was performed as an automated process using the Janus G3 workstation (Perkin Elmer, Turku, Finland). 

PCR Master Mix was prepared and pipetted in 384-well format. The extracted DNA was added to the PCR Master Mix and the plate was then sealed and run on a Quantstudio7DX or Viia7 RealtimePCR machine (Thermo Fisher, Massachusetts, USA). 

### 2.4. Data Analysis and Cut-Offs

TREC concentrations were calculated based on ∆Ct values between TREC and the RPP30 reference gene and TREC concentration unit was copies per 10^5^ cells. The following formula for calculation was provided by the manufacturer and used for the validation study:∆Ct = TREC Ct − RPP30 Ct
TREC: 2 × 2^−∆Ct^ × 117,000

The manufacturer of the EONIS kit later changed the formula to also adjust for the median population RPP30 Ct. The current formula is as follows:∆Ct =TREC Ct − RPP30 med Ct − (0.25 × (RPP30 Ct − RPP30 med Ct))
TREC: 2 × 2^−∆Ct^ × 117,000
where RPP30 med Ct is the median population RPP30 Ct value as provide by the kit manufacturer.

Detailed information on the formula can be found in Gutierrez-Mateo et al. [7].

Cut-offs for the neonatal screening program were determined based on the results of the validation study. The cut-off was set at a level where classical SCID newborns with complete T-cell depletion would be detected, but most T-cell lymphopenias with reduced amount of TRECs would not be reported as screen positives. 

### 2.5. Screening Algorithm

The screening algorithm is displayed in Figure 1. All neonatal samples were initially run as singletons. If TREC value of the sample was below cut-off for the initial analysis, samples were re-run in duplicates. If TREC values of the duplicate run were below the re-test cut-off, samples were reported as SCID screen positive. Reporting of screen-positive infants differed based on gestational age of the infant. If the child was born before week 32, we requested a repeated sample when the child reached an age equivalent to week 32. If the child was born between week 32 and 35, we requested a repeated sample two weeks after the initial sample. If the child was born after week 35 or the repeated sample remained SCID screen positive, a specialized hospital unit responsible for the follow-up and treatment of all SCID screen positive newborns was contacted. The hospital unit was then responsible for contacting the parents of the newborn as well as performing additional diagnostic tests, including flow cytometry and next generation sequencing with an immune-disorder-specific panel.

## 3. Results 

### 3.1. Validation Study

More than 6000 archived neonatal DBS samples were included and analyzed in the validation study. The TREC concentration distribution for a newborn population was determined and the median TREC concentration was 2422 copies per 10^5^ cells and the interquartile range was 1566–3718 copies per 10^5^ cells. 

Of the 17 SCID-positive samples, only two had measurable TREC levels, and only in one of three runs. TREC concentrations of these samples were 10 copies per 10^5^ cells and 40 copies per 10^5^ cells, respectively. Based on these results, we decided a cut-off of 50 copies per 10^5^ cells. When we initiated the screening, the cut-off was the same for the initial and re-test runs. This was later altered to 100 copies per 10^5^ cells for the initial run and 50 copies per 10^5^ cells for the re-test run. The differentiated cut-offs for the initial and re-test phases were introduced to ensure that screen positive samples would not be missed due to assay variation. 

We calculated intra and inter assay variation for a subset of the samples. The median intra assay CV was 14.5% with a range between 4.1% and 46.9%. The median inter assay CV was 16.6% with a range between 7.7% and 69.9%.

### 3.2. Screening

A total of 53.221 newborn screening DBS samples were analyzed between 1st February 2020 and 30th November 2020. The distribution of the TREC concentration stratified by month is shown in Figure 2. The overall sample median for the screening period was 1800 copies per 10^5^ cells and the interquartile range was 1263–2456 copies per 10^5^ cells.

We identified one screen-positive newborn in the first 10 months of screening and, thus, had a positive rate of approximately 1:54,000. The infant was born at term and had no measurable levels of TRECs. The infant did not have classic SCID but was diagnosed with a syndrome where the lack of mature T-cells is part of the syndrome. 

We also identified one preterm infant with a screen-positive DBS sample, resulting in a re-test rate of approximately 1:54,000. The premature baby was born at week 28. TREC values were 30 copies per 10^5^ cells. The infant had a repeated test preformed 4 weeks after birth and this repeated sample was screen negative. 

In general, the assay performed well, with analysis re-run rates of 0.25% primarily due to sub optimal DNA extraction.

## 4. Discussion

Screening for SCID was implemented in the Danish Newborn Screening Program 1 February 2020. During the first 10 months, we screened approximately 54,000 newborns and among these we identified one screen-positive infant and one premature infant with an initial screen-positive result, but with a screen-negative result on the repeated sample. The incidence of SCID was approximately 1:50,000; however, some variation between countries have been reported [8]. In this paper, we only presented data from the first 54,000 samples so we did not have sufficient data to make a valid estimate of the SCID incidence in Danish screening. A known issue in SCID screening is the false-positive screening results in preterm newborns [9,10]. We only identified one preterm newborn with a positive initial SCID screening. This low number of screen-positive preterm newborns might be due to our screening cut-off. As we are among the first screening programs to use the EONIS kit and report TREC concentrations as copies per 10^5^ cells, it is difficult to compare cut-off settings with that of other screening programs. If we stratify Ct values for TREC and RPP30 by gestational age, we find that infants born before week 28 had higher Ct values for TREC and lower Ct values for RPP30 compared to infants born at week 28 or later. The higher Ct value for TREC and the lower Ct values for RPP30 should result in an increased likelihood of a screen positive result. As we observed higher TREC Ct values in very preterm infants, our finding is likely to be in line with other screening programs, and the lack of premature screen-positive infants was likely caused by our cut-off level.

We found slightly lower concentrations of TREC per 10^5^ cells in fresh DBS samples compared to archived samples. This might be an effect of storage time. We found a slightly higher median RPP30 Ct value in the archived samples, median RPP30 Ct = 24.0 compared to median RPP30 Ct = 23.0 in samples from the first 10 months of screening. In terms of TREC Ct values, the archived samples had a median TREC Ct of 30.5 compared to 30.8 in the fresh DBS samples. This could indicate that TRECS are more stable than the reference gene, but further investigations should be performed and repeated analyses of the samples after storage would be beneficial. The differences in TREC concentrations did not affect the specificity of the screening, as this did not result in multiple false-positive samples. The cut-offs were initially based on the results of the validation study, and the results from the first 10 months of screening confirm that the current cut-off level was suitable for the screening. 

## Figures and Tables

**Figure 1 IJNS-07-00054-f001:**
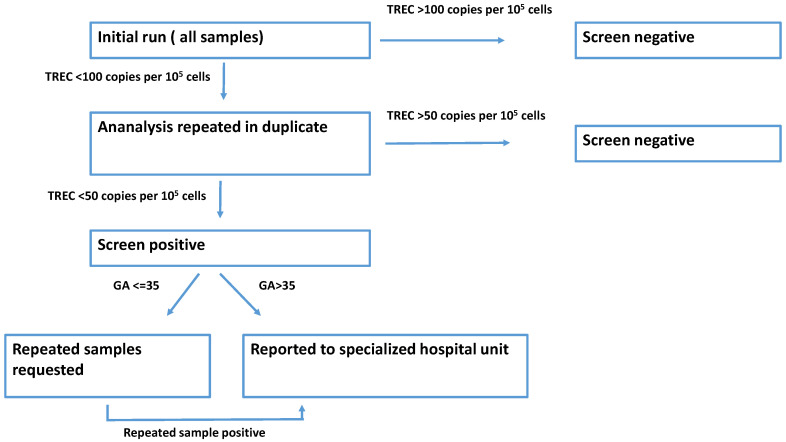
Screening algorithm for SCID screening.

**Figure 2 IJNS-07-00054-f002:**
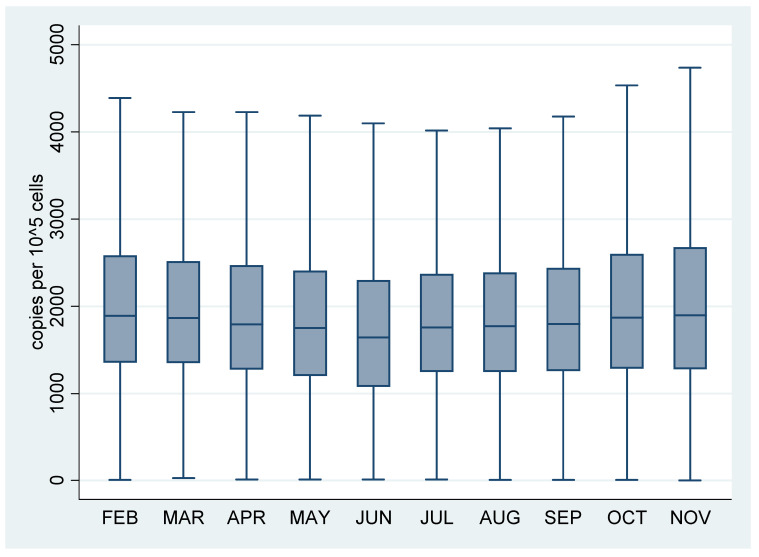
Boxplot of TREC values for all samples analyzed between February 2020 and November 2020 stratified by month. TREC values in copies per 10^5^ cells. Boxes represent median and interquartile range.

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
