# Peer review of "Implementation of SCID Screening in Denmark"

_2409-515X, 2021, doi:10.3390/ijns7030054_

Round 1
Reviewer 1 Report
- Only in the Results section the authors mention that 53.221 newborns were tested. I would suggest to mentioning that in the Introduction as well, because the number of cards tested in the validation phase was also mentioned.
- The validation study was discussed first in the results section followed by the prospective inclusion of 53.221 newborns. I would stick to the same order in the methods section
- The infant identified with low TRECs did not have classic SCID but was diagnosed with a syndrome where lack of mature T cells is part of the syndrome. Maybe some more detail can be given.
- in section 2.4 it is good to refer to a paper in which the calculation is published, however, for the readibility it would help to give some information, e.g. In short, the cut-off value was determined.......
- I would recommend visualizing the screening algorithm in a scheme. If only 1 figure is allowed it can be added as A or B panel.
- Lower TRECs in fresh sample: "This might be an effect of storage time". I would recommend that the author make a proposal how this can be tested to confirm. Is this also seen in other programs? TRECs are very stable circles, so maybe the DNA isolation is more efficient or the circle is more stable than the reference gene (embedded in genomic DNA). This needs some more discussion and maybe reference to other studies if available.
Author Response
Reviewer1:
Comments and Suggestions for Authors
- Only in the Results section the authors mention that 53.221 newborns were tested. I would suggest to mentioning that in the Introduction as well, because the number of cards tested in the validation phase was also mentioned.
Response:
The number of screened newborn has now been added to the introduction.
- The validation study was discussed first in the results section followed by the prospective inclusion of 53.221 newborns. I would stick to the same order in the methods section
Response:
The order has now been changed in the methods section
- The infant identified with low TRECs did not have classic SCID but was diagnosed with a syndrome where lack of mature T cells is part of the syndrome. Maybe some more detail can be given.
Response:
We agree with the reviewer that more details would be beneficial. As we do not have permission from the infant’s parents to share the medical information, we are not allowed to go into further details.
- in section 2.4 it is good to refer to a paper in which the calculation is published, however, for the readibility it would help to give some information, e.g. In short, the cut-off value was determined.......
Response:
We agree with the reviewer and we have now added the formula for calculating TREC concentrations to the text, and refer to a paper for further details on how the formula originated.
- I would recommend visualizing the screening algorithm in a scheme. If only 1 figure is allowed it can be added as A or B panel.
Response:
We have now added a figure displaying the screening algorithm.
- Lower TRECs in fresh sample: "This might be an effect of storage time". I would recommend that the author make a proposal how this can be tested to confirm. Is this also seen in other programs? TRECs are very stable circles, so maybe the DNA isolation is more efficient or the circle is more stable than the reference gene (embedded in genomic DNA). This needs some more discussion and maybe reference to other studies if available.
Response:
We thank the reviewer for the comment and suggestion to expand the discussion. We have now added data on the Ct values of TREC and RPP30 from the archived samples in the validation study and from the first 10 months of screening. We do find higher RPP30 Ct values in the archived samples, but similar TREC CT values compared to the fresh samples. This could indicate that TRECs are more stable than RPP30, however it might also be kit lot specific effects that we have not accounted for. We also suggest re-analyzing our screening samples after longtime storage.
Reviewer 2 Report
Bækvad-Hansen et al. present short-term data on the implementation of TREC newborn screening in Denmark. The data is of general interest to the newborn screening community, but could be easily improved as follows.
Major revisions:
- Why was the cut-off for the initial run altered?
- The syndrome identified by TREC screening should be named.
- The number of detected preterm infants is surprisingly low, as compared to other screening programs. The authors should comment on that.
- The SCID incidence established by TREC screening is approx. 1.50,000. The authors should discuss their data considering that.
Minor revisions:
- In the introduction all information should be accompanied by references.
- There is functional SCID, e.g., MALT1-deficiency, with normal T-cell counts and normal TRECS. The introduction should briefly comment on that.
- The capillaries used to collect samples should be specified, because it is of special interest that they are not containing anticoagulants.
- The realtime PCR device used should be specified.
Author Response
Bækvad-Hansen et al. present short-term data on the implementation of TREC newborn screening in Denmark. The data is of general interest to the newborn screening community, but could be easily improved as follows.
Major revisions:
- Why was the cut-off for the initial run altered?
Response:
The differentiated cut-offs for initial and re-test phase was introduced to ensure that screen positive samples would not be missed do to assay variation. We have now added this sentence to the manuscript.
- The syndrome identified by TREC screening should be named.
Response:
We agree with the reviewer that the manuscript would benefit from naming the syndrome. As we do not have permission from the infant’s parents to share the medical information, we are unfortunately not allowed to do so
- The number of detected preterm infants is surprisingly low, as compared to other screening programs. The authors should comment on that.
Response:
Thank you for pointing this out. We do find very few screen positive preterm infants, even if we look at all screening results from February 1st 2021 to now. We have added a paragraph in the discussion section about this issue. As the Eonis kit is so new, and the calculation and reporting of TREC concentrations is so different from what most other countries use, it is difficult to compare our cut-off values to others. The low number of screen positive infants is most likely due to our low cut-off. We are to find SCID positives, but not other t-cell lymphopenias and the cut-offs have been set to accommodate this.
- The SCID incidence established by TREC screening is approx. 1:50,000. The authors should discuss their data considering that.
Response;
Thank you for this important comment. We have now added a paragraph addressing the issue
Minor revisions:
- In the introduction all information should be accompanied by references.
Response:
We have added more reference to the text.
- There is functional SCID, e.g., MALT1-deficiency, with normal T-cell counts and normal TRECS. The introduction should briefly comment on that.
Response:
The reviewer is correct, that the screening will only detect SCID with no or very low T-cell counts. This limitation is now addressed in the introduction.
- The capillaries used to collect samples should be specified, because it is of special interest that they are not containing anticoagulants.
Response:
We have added information about the sampling, including the specifications for use of capillary tubes.
- The realtime PCR device used should be specified.
Response:
We have added the name of the Realtime PCR device and the manufacturer to the text.
Reviewer 3 Report
Bækvad-Hansen, Marie’s paper summarizes perfectly the validation and implementation of SCID screening in Denmark, as well as the most important results obtained. In order to offer more complete information in some aspects and help other readers of the article to compare results obtained in other NBS programs for SCID I make some recommendations as follows. I recommend to the editor the publication of this article, which provides very new results in the field of newborn screening for SCID with new methodologies.
INTRODUCTION
- It would be appropriates to know what are the other diseases included in the Danish NBS Program.
- When the author says “Screening for SCID is based on quantification of T cell receptor excision circles (TREC)” please add “in DBS samples”.
MATERIALS AND METHODS
2.3. Realtime PCR
- Please, specify if Eonis DNA extraction kit was used by manual of automated process. If it was by automated process, please specify the instrument.
- Please, specify Realtime PCR instrument.
- Please, add a “dot” in the end of the paragraph.
2.4. Data analysis and cut-offs
- Please specify in this paragraph how you wanted to establish the cutoffs.
- Please, review the spelling of the cite (Gutierrez-Mateo et al.)
2.5. Screening algorithm
- You should explain what it is your decision depending of the results obtained in the second samples. For example: “If the child was born after week 35, or the second sample is still below the cutoff in the other two cases mentioned, a specialized hospital unit responsible for the follow-up…”
- Have a figure with the decision algorithm and cutoffs of retest, requests 2nd sample and positive detection would help the reader of the paper.
- Please consider add what are the additional diagnostic tests use to discard/confirm SCID cases or reference bibliography.
RESULTS
3.1. Validation study
Please, specify if you perform LD, LQ, and both intra and inter-serial variability for the validation study. This is important in order to establish your cutoffs.
3.2. Screening
Please consider add retest rate, second sample rate and positive detection rate, it would be informative for the reader and to compare with the effectiveness of other SCID NBS programs.
Please explain to the reviewer why figure 1 has been included and what information does it provide.
A suggestion: please, in the case for preterm newborns, reconsider to call only positive detection to the cases with low TREC values in second samples requested, since they will be the cases that will have to be referred to the hospital. But I know this decision depends on each NBS program.
- Please, provide more information about what syndrome was found in the second screen positive newborn. Finally, it was classified as an immunodeficiency-non SCID?.
Author Response
Bækvad-Hansen, Marie’s paper summarizes perfectly the validation and implementation of SCID screening in Denmark, as well as the most important results obtained. In order to offer more complete information in some aspects and help other readers of the article to compare results obtained in other NBS programs for SCID I make some recommendations as follows. I recommend to the editor the publication of this article, which provides very new results in the field of newborn screening for SCID with new methodologies.
INTRODUCTION
- It would be appropriates to know what are the other diseases included in the Danish NBS Program.
Response:
We feel that naming all the disorders in the Danish NBS Program would take up too much space in a limited text. We have instead listed the group of disorders covered by the screening and included a link to the program, where all disorders are listed.
- When the author says “Screening for SCID is based on quantification of T cell receptor excision circles (TREC)” please add “in DBS samples”.
Response:
This has now been added to the text.
MATERIALS AND METHODS
2.3. Realtime PCR
- Please, specify if Eonis DNA extraction kit was used by manual of automated process. If it was by automated process, please specify the instrument.
Response:
The DNA extraction was performed by automated process. This has now been added to the text.
- Please, specify Realtime PCR instrument.
Response:
We have added the name and manufacturer of the Realtime PCR instrument to the text.
- Please, add a “dot” in the end of the paragraph.
Response:
We have added a “dot” in the end of the paragraph
2.4. Data analysis and cut-offs
- Please specify in this paragraph how you wanted to establish the cutoffs.
Response:
We agree with the reviwer, that information about cut-offs is too sparse. We have now added information about the cut-offs settings to the section.
- Please, review the spelling of the cite (Gutierrez-Mateo et al.)
Response:
The spelling error has now been corrected
2.5. Screening algorithm
- You should explain what it is your decision depending of the results obtained in the second samples. For example: “If the child was born after week 35, or the second sample is still below the cutoff in the other two cases mentioned, a specialized hospital unit responsible for the follow-up…”
Response:
The reviewer has a very valid point. We have now added to the text, that children born after week 35 ot infants with repeated screen positive results are reported to a specialized hospital unit for diagnostic follow-up.
- Have a figure with the decision algorithm and cutoffs of retest, requests 2nd sample and positive detection would help the reader of the paper.
Response
Thank you for the suggestion. We have now added an additional figure displaying the screening algorith
- Please consider add what are the additional diagnostic tests use to discard/confirm SCID cases or reference bibliography.
Response:
We have now added information about the additional diagnostic testing to section 2.5
RESULTS
3.1. Validation study
Please, specify if you perform LD, LQ, and both intra and inter-serial variability for the validation study. This is important in order to establish your cutoffs.
Response:
Thank you for the questions regarding limits and variation. In regard to limits of detection and limits of quantification, this is not something that is provided from the manufacture and we have not calculated these. We do however include low level controls in each run, and know, that the assay can distinguish between very low levels of TRECS and no TRECs. In terms of determining cut-offs, this was based on the results of the SCID positive samples. As only two SCID positive samples had measurable TRECs in one of their triplicate analyses, it is difficult to estimate variance for the SCID positive samples and we instead set a limit that would ensure us to have caught all of the SCID positive cases in the validation study. In regard to variation of the non SCID positive samples, we found a intra assay CV % of 14.5% (range 4.1% -46.9%) and a inter assay CV % of 16.6% ( range 7.4% -69.6%). For the general quality assessment of a run, we have multiple parameters in regard to Ct values of the reference gene and ∆Ct values for the high level TREC control that needs to be fulfilled in order for a run to be valid.
We have added these information to the manuscript
3.2. Screening
Please consider add retest rate, second sample rate and positive detection rate, it would be informative for the reader and to compare with the effectiveness of other SCID NBS programs.
Response:
We have now added the rates to the results section
Please explain to the reviewer why figure 1 has been included and what information does it provide.
Response:
The Eonis kit is new and is the first to report TREC concentrations as copies per 105 cells. Figure 1 (now Figure2) is included to provide information on the range of TREC concentrations for DBS samples as well as the variability of the assay.
A suggestion: please, in the case for preterm newborns, reconsider to call only positive detection to the cases with low TREC values in second samples requested, since they will be the cases that will have to be referred to the hospital. But I know this decision depends on each NBS program.
Response:
This is a very valid point. In the manuscript, we now say, that we find one screen positive infant and one preterm infant with an initial screen positive sample, but a screen negative repeated sample. This preterm is not part of our positive detection rate.
- Please, provide more information about what syndrome was found in the second screen positive newborn. Finally, it was classified as an immunodeficiency-non SCID?.
Response:
We agree with the reviewer that the manuscript would benefit from naming the syndrome. As we do not have permission from the infant’s parents to share the medical information, we are unfortunately not allowed to do so
Round 2
Reviewer 2 Report
The authors have addressed my reviews thoroughly.